# The ABC of Immune-Mediated Hepatitis during Immunotherapy in Patients with Cancer: From Pathogenesis to Multidisciplinary Management

**DOI:** 10.3390/cancers16040795

**Published:** 2024-02-15

**Authors:** Angioletta Lasagna, Paolo Sacchi

**Affiliations:** 1Medical Oncology Unit, Fondazione IRCCS Policlinico San Matteo, 27100 Pavia, Italy; 2Division of Infectious Diseases I, Fondazione IRCCS Policlinico San Matteo, 27100 Pavia, Italy; p.sacchi@smatteo.pv.it

**Keywords:** hepatitis, immune checkpoint inhibitors, immune-related adverse events (irAEs), CD8+ T cells, NAFLD, drug-induced liver injury (DILI), anti-CTLA4 antibodies, anti-PD-1, anti-PD-L1, viral hepatitis

## Abstract

**Simple Summary:**

Immune-mediated hepatotoxicity (IMH) is a not-so-rare complication of immune checkpoint inhibitors (ICIs) and it is extremely heterogeneous in its clinical presentation and severity. This narrative review aims to report the current knowledge on hepatic immune-related adverse events (irAEs) during immunotherapy from pathogenesis to multidisciplinary management.

**Abstract:**

Immune-mediated hepatotoxicity (IMH) is not-so-rare complication during treatment with immune checkpoint inhibitors (ICIs). This narrative review aims to report the current knowledge on hepatic immune-related adverse events (irAEs) during immunotherapy from pathogenesis to multidisciplinary management. The majority of cases of IMH are asymptomatic and only a few patients may have clinical conditions. The severity of IMH is usually stratified according to Common Terminology for Clinical Adverse Events (CTCAE) criteria, but these scores may overestimate the clinical severity of IMH compared to the Drug-Induced Liver Injury Network (DILIN) scale. The differential diagnosis of IMH is challenging because the elevated liver enzymes can be due to a number of etiologies such as viral infection, autoimmune and metabolic diseases, liver metastases, biliary diseases, and other drugs. The cornerstones of IMH management are represented by withholding or delaying ICI administration and starting immunosuppressive therapy. A multidisciplinary team, including oncologists, hepatologists, internists, and emergency medicine physicians, is essential for the management of IMH.

## 1. Introduction

Over the last 15 years, immune checkpoint inhibitors (ICIs) have become promising therapies for patients with solid tumors [1]. Ipilimumab was the first checkpoint inhibitor approved by the FDA in 2011 for the treatment of advanced melanoma. It is a cytotoxic T-lymphocyte antigen number 4 (CTLA-4)-blocking antibody. There are three main immune checkpoint molecules approved for the treatment of solid tumors: (1) anti-programmed cell death protein (PD)-1 agents (i.e., nivolumab, pembrolizumab, and cemiplimab), (2) anti-PD ligand (L)-1 agents (i.e., atezolizumab, avelumab, and durvalumab), and (3) anti-CTLA-4 agents (i.e., ipilimumab and tremelimumab) [2]. ICIs enhance the anti-cancer immune response by blocking inhibitory receptors on the T-cell membrane and reversing T-cell exhaustion. They significantly improve patient survival in several cancer types, such as melanoma, kidney, lung, breast, and uroepithelial cancer [3]. However, as a result of their specific mechanism of action, ICIs can cause immune-related adverse events (irAEs) due to a loss of self-tolerance [4,5]. IrAEs can virtually affect all organs and multiple presentations of irAEs have been reported. They present various clinical manifestations, types of onset, kinetics, and severity and so they require specific multidisciplinary management [6,7]. The most common irAEs are dermatological, gastrointestinal, hepatic, pulmonary, and endocrinological toxicities [8]. Hepatotoxicity is quite common during treatment with ICIs, typically being mild to moderate, but may lead to liver failure and death [9].

This narrative review aims to report the current knowledge on hepatic irAEs during immunotherapy from pathogenesis to multidisciplinary management.

## 2. Epidemiology

### 2.1. Incidence

The real incidence and severity of immune-mediated hepatic adverse events are challenging due to extreme variability and the discrepancy between data extrapolated from clinical trials and real-world cohort studies. In a randomized, double-blind, multicenter, phase 2, dose-ranging study of ipilimumab for advanced melanoma, only two patients assigned the 10 mg/kg dose reported liver adverse events ≥ grade 3 (2/71, 3%), while no liver adverse events were recorded in patients allocated the 0·3 mg/kg dose and the 3 mg/kg dose [10]. Also, in the randomized, open-label, international, phase 3 study about the efficacy and safety of nivolumab in patients with advanced squamous-cell non–small-cell lung cancer (NSCLC), no liver adverse events ≥ grade 3 were reported [11]. Shortly thereafter, an increasing number of hepatic irAEs in real-world settings were described. De Martin et al. reported acute mixed (hepatocellular and cholestatic) hepatitis in 19 cancer patients (3.5%) with ICIs at Institut Gustave Roussy (Villejuif, France) [12]. Parlati and colleagues described an adjusted risk of 2.0 of grade 3 or more liver function test abnormality under ICIs (1.5–2.6, 95%CI [confidence interval]) [13]. Recently, a pharmacovigilance study based on data from the FAERS database has analyzed 1481 reports of immune-mediated liver toxicity. The authors have demonstrated that ICIs are significantly associated with liver failure (IC = 1.24, 95% CI, 1.06–1.42) and that ICI combination therapy presents a greater risk compared to ICI monotherapy [14].

### 2.2. Risk Factors

The reported incidence of immune-mediated hepatotoxicity (IMH) varies according to the class of ICI, the type of tumors, and underlying clinical conditions. In the above-mentioned pharmacovigilance study, the authors highlighted that anti-PD-L1 treatment has a risk of 1.20-fold and 1.31-fold greater than those with anti-CTLA-4 and anti-PD-1 treatments, respectively [14]. This observation is in contrast with a meta-analysis of 17 clinical trials that described a higher risk of hepatotoxicity in the case of treatment with anti-CTLA-4 than with anti-PD-1 (odds ratio [OR] for high-grade hepatoxicity = 1.52 vs. 0.48) [15]. The incidence of liver toxicity with anti-CTLA-4 monotherapy ranges from 2% to 15%, while with anti-PD-1 and anti-PD-L1 it ranges from 0% to 3% and 0% to 6%, respectively [16]. ICI combination therapy showed a greater rate of hepatotoxicity compared with ICI monotherapy (IC = 0.63, 95% CI, 0.99–2.04; ROR = 1.31 [95% CI, 1.80–2.31]) [14]. In the phase III clinical trial Checkmate 067, patients with advanced melanoma treated with ipilimumab plus nivolumab presented a higher alanine aminotransferase (ALT) increase (19%) when compared to those receiving ipilimumab (4%) or nivolumab (4%), while the increase in aspartate aminotransferase (AST) was 17%, 4%, and 4%, respectively [17]. Additionally, previous ICI treatment is an independent risk factor for IMH [2].

Concerning the type of tumors, patients with hepatocellular carcinoma (HCC) and concomitant chronic hepatitis or cirrhosis seem to have a higher risk of ALT increase with non-liver cancers [18]. The risk of IMH has been reported to be associated in particular with patients with melanoma (OR= 11.6, *p* = 0.002) [2]. Moreover, a meta-analysis showed that the presence of liver metastases seems to be not a risk factor for developing IMH (OR: 1.468, 95% CI: 0.991 to 2.175, *p* = 0.056) [19].

As regards HLA antigen expression, a recent study demonstrated a strong correlation between immune-related pneumonitis and germinal expression of HLA-B*35 and DRB1*11, both alleles associated with autoimmune diseases [20]. Other HLA alleles such as DRB1*0301, DRB3*0101, DRB1*0401, and DRB1*07 have been associated with iAIH risk, but no association of these alleles has been related with IMH to date [21].

Patients with preexisting chronic liver disease, such as non-alcoholic fatty liver disease (NAFLD), showed a higher incidence of IMH than those without. It is probably due to the generation of more free radicals in hepatocytes with the enhanced production of neoantigens that leads to liver immune aggression [22].

A risk factor associated with IMH was the onset of fever above 38 °C within 24 h after the start of the initial ICI treatment (hazard ratio [HR] = 6.21, *p* < 0.001) [23], with more discordant evidence about female sex as a risk factor for IMH [19].

Our recent work has demonstrated that chronic proton pump inhibitor (PPI) use is associated with an increased risk of GI irAES, including IMH (HR 13.22, 95% CI 3.11–56.10, *p* < 0.000). This HR was confirmed after weighing for the propensity score (HR 15.13 95% CI 3.22–71.03, *p* < 0.000) [24].

## 3. Pathogenesis

### 3.1. The Liver as an Immunological Organ

The liver presents a unique immunological profile: since it is constantly exposed to foreign antigens through the intestinal and systemic circulations, immune tolerance to non-pathogenic antigens is essential to avoid chronic inflammation [25]. The liver is the only non-lymphoid organ to induce primary activation of naïve CD8+ T cells [26]. This activation can lead at the same time to fully functional and effective CD8+ T cells and to ineffective activation of CD8+ T cells with impaired cytotoxic capacities and a reduced half-life [27]. The liver is made of parenchymal cells (hepatocytes and cholangiocytes), liver sinusoidal endothelial cells (LSECs), hepatic stellate cells (HSCs), and a complex immune cell network of myeloid and lymphoid cells. Sinusoids are lined by LSECs and provide anchoring places for cells of the immune system [28]. The liver is exposed to bacterial lipopolysaccharides (LPS) from the intestinal flora with consequences for the development of the immune response: “endotoxin tolerance”. This term corresponds to the loss of stimulation by TLR4 following the exposure of cells to low-LPS (the natural ligand of TLR4) concentrations. Moreover, exposure to LPS leads to the establishment of an immunological microenvironment with the production of IL-10, TGF-b, HGF (hepatocyte growth factor), and retinoic acid by stellate-Ito cells [29]. At the end of an immune response, only a small population of memory T cells remains. This removal of CD8+ T cells from the periphery is linked to the accumulation of apoptotic CD8+ T cells in the liver [30]. The constitutive expression of intercellular adhesion molecule 1 (ICAM-1) and vascular cell adhesion molecule 1 (VCAM-1) is induced by LSECs and Kupffer cells in the hepatic sinusoids, allowing entry into the parenchyma of activated CD8+ T cells [31]. These activated CD8+ T cells interact with Kupfer cells via their FasL molecules and the secretion of IFN-gamma stimulates the production of TNF-alfa. A large amount of TNF-alfa can be the cause of hepatocyte death, the so-called “bystander hepatitis phenomenon”, which is the pathogenetic mechanism observed after infection by non-hepatotropic viruses, such as influenza and EBV that activate a large number of T cells [32].

### 3.2. Putative Mechanisms of Liver Damage during ICIs

PD-L1 is expressed on non-parenchymal liver cells and may be induced on hepatocytes during inflammation. The expression of PD-L1 together with the presence of CTLA-4+ and CD4+ Tregs, promotes tolerance to self-antigens within the liver by downregulating effector T cells. The perturbation of these pathways through ICIs may lead to an immune-mediated injury [25]. Different mechanisms may be responsible for IMH such as: (i) increased T-cell function, concomitant Tregs depletion, and the consequent reduction in anti-inflammatory cytokines such as interleukin (IL)-10, IL-35, and TGF-β with the modulation of the interaction between adaptive-innate immunity; (ii) enhanced humoral autoimmunity, with the enhancement of preexisting auto-antibodies; (iii) direct effect via complement-mediated injury; (iv) the expansion of T helper cells, such as the Th1 and Th17 cells, with the increase in the levels of proinflammatory cytokine (IL-2, IFN-γ, and TNF) production, which can go on to activate cytotoxic T lymphocytes [6,33,34]. Finally, as mentioned above, the expression of ICAM-1 and VCAM-1 promotes the interaction of activated CD8+ T cells in the systemic circulation with Kuffer and LSECs, leading to the retention of activated CD8+ T cells in the liver. The secretion of TNF-α secretion by Kuffer cells and hepatocyte injury [35]. Despite all these mechanisms, we are still unable to establish why only some patients develop IMH and others do not.

## 4. Diagnosis

### 4.1. Clinical Presentation

The majority of IMHs are asymptomatic and only a few patients may have clinical conditions such as fatigue, abdominal discomfort, fever, rash, and jaundice [36]. Fever is more prevalent in IMHs induced by anti-CTLA-4 inhibitors [12]. The liver imaging in mild to moderate IMH is usually normal, while mild portal lymphadenopathy, periportal edema, and hepatomegaly may be seen in severe IMH [37]. The severity of IMH is usually stratified according to the Common Terminology for Clinical Adverse Events (CTCAE) criteria, but these scores may overestimate the clinical severity of IMH compared to the Drug-Induced Liver Injury Network (DILIN) criteria [38].

The classification of the severity of liver damage should be considered beyond ALT elevation, as should the impairment of the International Normalized Ratio as occurs in the DILIN system which also considers symptoms and other organ failure indexes [39]. However, we do not have an explicit criterion for IMH grading different from elevated liver function tests and we do not know which of them is more useful in predicting the prognosis of IMH. (Table 1).

### 4.2. Differential Diagnosis

The differential diagnosis of IMH is challenging because the elevated liver enzymes can be due to many etiologies such as viral infection, autoimmune and metabolic diseases, liver metastases, biliary diseases, and other drugs. Therefore, it is mandatory to rule out the other causes of liver injury [40].

Firstly, we should exclude hepatic virus infection such as HAV, HBV, HCV, HEV CMV, EBV, and HSV using the related bioumoral tests (anti-HAV IgM, HBsAg, anti-HBc IgG, anti-HBc IgM, HBV DNA, anti-HCV, HCV RNA, anti-HEV IgG, anti-HEV IgM, HEV RNA, anti-CMV IgM, CMV DNA, anti-EBV IgM, EBV DNA, anti-HSV IgM, and HSV DNA). The research of anti-tissue antibodies (ANA, ASMA, anti-LKM-1, anti-LC-1, anti-SLA/LP, pANCA, and serum IgG, IgM, IgA) is recommended to rule out autoimmune diseases.

The differential diagnosis between ICI-induced hepatitis and drug-induced autoimmune liver disease is particularly challenging because there are at least seven phenotypes of this form.

The first one, AIH with DILI occurs in patients with known quiescent AIH, and the drug acts as a trigger of the autoimmune liver disorder. In this case, histology may show advanced fibrosis. The second is drug-induced—AIH that affects patients with a low-grade misdiagnosed disease where a drug produces an immune reaction leading to a chronic process and permanent need for immunosuppressive therapy. Other less common forms are IM-DILI, due to autoimmune hypersensitivity frequently presenting with systemic symptoms, indistinguishable from true AIH and responding to IS treatment, mixed autoimmune type, often associated with other autoimmune disorders, and DILI-positive autoantibodies.

Hepatic imaging (ultrasonography, CT scan, and MRI) can be useful to exclude the progression of cancer. The medication history can help to evaluate chemotherapeutic drugs that may also cause liver injury (dacarbazine, carboplatin, and complementary and herbal medication, or alcohol abuse). NAFLD is due to lipid accumulation in liver cells and tissue inflammation with cytokine release [41]. It can be a potential risk factor for IMH but also a differential diagnosis [42].

Finally, myocarditis, myositis, Wilson’s disease, and bone metastasis have to be excluded [43].

### 4.3. Histological Diagnosis

IMH is typically a clinical diagnosis and a liver biopsy is required only in the case of absence of response to steroids. The pathological features of IMH are scarce. There are clinicopathologic features that support the idea that IMH is a distinct entity from autoimmune hepatitis. IMH rarely presents autoantibodies and/or IgG elevations and the histological patterns are different from those seen in classical autoimmune hepatitis. The typical patterns include acute hepatitis with spotty or confluent necrosis in the centrilobular zone, and granulomatous hepatitis [44,45]. There may be acute hepatitis with lobular inflammation and acidophil bodies [46]. Lobular hepatitis has the same features as autoimmune hepatitis and with patterns of zone 3 panlobular inflammation. IMH inflammatory infiltrate differs from that of autoimmune hepatitis because it is predominantly composed of activated CD3+ and CD8+ T lymphocytes while CD20+ and CD4+ are characteristic of autoimmune forms [46,47]. In IMH caused by anti-CTLA-4, a pattern of granulomatous hepatitis, with fibrin-ring granulomas and central-vein endothelitis is found [12]. Mild portal fibrosis seems to suggest a trend of acute hepatitis towards chronicity [12].

The guidelines of the European Society of Medical Oncology recommend liver biopsy for patients with IMH and poor or slow response to steroids [48]. Liver biopsy may be useful in different settings: To exclude pre-existing diseases;To confirm the presence of a pattern consistent with liver injury in the case of ring granulomas and/or endothelitis observed in patients treated with anti-CTLA-4;To look for the distinctive picture of liver toxicity related to anti-PD-1/PD-L1 and anti-CTLA4;To estimate the degree of liver injury;To distinguish IMH from AIH with an atypical clinical pattern;To assess a possible evolution to a chronic form of liver damage;Liver biopsy has also several limitations, such as cost and the possibility of adverse events, thus should be considered only if it may change the management of patients.

Li and colleagues conducted a retrospective analysis of liver biopsy in a cohort of 213 patients who developed IMH. In this paper, patients with ICI hepatitis undergoing a liver biopsy showed longer median (IQR) time to ALT normalization (42 [6,32,33,34,35,36,37,38,39,40,41,42,43,44,45,46,47,48,49,50,51,52,53] vs. 33 [6,27,28,29,30,31,32,33,34,35,36,37] days; *p* = 0.01) and to ALT levels of 100 U/L or less (21 [2,17,18,19,20,21,22,23,24,25] vs. 15 [14,15,16,17] days; *p* = 0.01). These data underline that performing a liver biopsy in this setting may delay the initiation of corticosteroids and in the resolution of liver inflammation [49].

Parlati et al. conducted a retrospective monocenter study to explore the impact of liver biopsy on the clinical management of IMH. The time for normalization of transaminases was 49 days in those patients who did not undergo liver biopsy and 60 days in the group of patients who did (*p* = 0.205) [50].

## 5. Management

### 5.1. General Recommendations

IMH is caused by an excessive immune response against liver tissue. Therefore, the cornerstones of IMH management are represented by withholding or delaying ICI administration and starting immunosuppressive therapy [51]. A multidisciplinary team, including oncologists, hepatologists, internists, and emergency medicine physicians, is essential for the management of IMH. The international guidelines recommend specialist gastroenterology/hepatology consultation in patients with high-grade IMH. The European Association for the Study of the Liver (EASL) suggests that a multidisciplinary team with a hepatologist should manage all “sufficiently severe” IMHs [52]. Patients with ICI-related hepatitis should be managed by a team including a hepatologist as suggested by the American Gastroenterological Association (AGA) [53], ASCO [54], and ESMO [48], and, above all, in cases of grade ≥ 3 liver enzyme elevation (AGA and high-grade hepatitis (ASCO)) ESMO suggests to refer to hepatologists only if the patient did not respond to a second line immunosuppressive agent.

Li and colleagues reported that early gastroenterology/hepatology consultation in patients with steroid-refractory disease was associated with faster ALT normalization (hazard ratio [HR], 1.89; 95% CI, 1.12–3.19; *p* = 5.017) and ALT improvement to ≤100 U/L (HR, 1.72; 95% CI, 1.04–2.84; *p* = 5.034). This beneficial effect was not evident in patients with steroid-responsive hepatitis (HR, 1.12; 95%, 0.83–1.51; *p* = 5.453) [55].

### 5.2. When to Start Corticosteroids?

Patients with any degree of ALT elevation should be screened with a full workup for other causes of viral involvement. Once ruled out other causes of liver diseases, management is according to the CTCAE (Table 1).

Grade 1 hepatitis can be managed with close monitoring of liver-associated enzymes while the patient continues ICIs. The Society for Immunotherapy of Cancer (SITC) recommends testing liver enzymes [56], while ASCO suggests checking liver enzymes twice a week [54].

Grade 2 hepatitis requires the withdrawal of ICIs. Liver enzymes should be monitored closely until they return to grade 1 level or normalize completely and if there is no improvement and/or increase upon repeat testing, oral corticosteroids (starting at 0.5–1 mg/kg/day of prednisone or an equivalent corticosteroid) can be given [48,54,56]. ICIs can be resumed once improvement is noticed, and corticosteroid reduction should occur slowly over time (4–12 weeks).

Grade 3 hepatitis imposes the permanent discontinuation of ICIs. Treatment with corticosteroids at higher doses (1–2 mg/kg/day (methyl)prednisolone) is recommended. If no improvement is seen in 3 days, physicians should consider adding a secondary agent such as mycophenolate mofetil [48,54,56].

Grade 4 hepatitis imposes the permanent discontinuation of ICIs with the start of corticosteroids at higher doses IV (2 mg/kg/day methylprednisolone) and the possibility of hospitalization. Intravenous steroids should be switched to an oral schedule and weaned in 4 weeks once ALT improves to grade 2 or less [48]. (Table 2).

If patients have to receive high-dose steroids for four weeks or longer, they are at risk of opportunistic infections (i.e., *Pneumocystis jiroveci*) and/or reactivation of chronic hepatitis B, so the expert panel agrees on the need to initiate prophylaxis [48,54]. To reduce the risk of infection, budesonide, a drug with 90% hepatic clearance and metabolism, may be evaluated in the case of IMH without liver failure. This type of corticosteroid has fewer side effects and could be maintained in the event of a resumption of ICIs to limit the recurrence of hepatitis and there is no need for dose tapering in order to reinitiate immunotherapy [57,58].

### 5.3. Refractory IMH to Steroid

Infliximab, a mouse–human chimeric monoclonal antibody anti-TNFα, should not be given to patients with hepatitis because infliximab carries a risk of hepatotoxicity [15]. If there is no improvement after high-dose steroids, a second immunomodulating agent (i.e., mycophenolate mofetil at the dosage of 500–1000 mg twice daily [12]; azathioprine 1–2 mg/kg/day [59], or tacrolimus with blood levels 8–10 ng/mL [60]) should be considered. The American Association for the Study of Liver Diseases (AASLD) guidelines recommend azathioprine as the first choice in case IMH is refractory to steroids, and mycophenolate mofetil in the case of a failure to respond to azathioprine [61]. No data demonstrate that azathioprine is superior to azathioprine in this setting [62]. Tacrolimus is a calcineurin inhibitor and can be used as a third-line option in patients not improving with mycophenolate mofetil [63]. Recently, plasma exchange has been used in patients with IMH non-responsive to steroids. Riveiro-Barciela and colleagues have reported clinical success in a woman after the failure of steroids and mycophenolate mofetil [64]. Tocilizumab is an anti-interleukin-6 and has been demonstrated to induce clinical improvement with just a single dose [65].

### 5.4. When to Rechallenge?

Current guidelines recommend permanent discontinuation of ICIs in the case of grade 4 IMH [48,54]. Several retrospective and few prospective studies described the risk of recurrence of irAEs after rechallenge. The rechallenge with ICIs may depend on the indication, the efficacy of the causative drug, the alternative therapeutic options, and the type and severity of IMH. There are three scenarios:(i)A class switch scenario from anti-PD-(L)1 to anti-CTLA-4 therapy or vice versa;(ii)A rechallenge scenario with reintroduction of the same molecule after resolution of IMH;(iii)A rechallenge scenario with the reintroduction of the same molecule concomitantly with immunosuppressive therapy.

A multicenter, retrospective cohort study at the Dana-Farber/Brigham and Women’s Cancer Center and the Massachusetts General Hospital Cancer Center evaluated the hepatitis recurrence after rechallenge. Of the 31 patients who resumed ICI therapy after IMH, six patients (19.4%) developed a new irAE. In particular, 4 patients (12.9%) developed recurrent IMH, 1 patient (3.2%) developed grade 2 pneumonitis, and 1 patient (3.2%) developed grade 3 hypophysitis [66].

Simonaggio and colleagues evaluated the safety of the rechallenge with anti-PD-1 or anti-PD-L1 after an irAE (17 patients have had hepatitis grades 2 to 4 [18%]). Three patients reported the same irAE after rechallenge, but the second irAE was not more severe than the initial IMH [67]. Pollack et al. described 29 patients with IMH (19 with ≥grade 3). Five of these 29 patients (17%) developed an irAE of ≥grade 3 after resuming therapy with anti-PD-1 [68].

The risk of liver injury after rechallenge depends on the class of ICIs. The reintroduction of an anti-CTLA-4 antibody in a patient with previous IMH due to an anti-PD-1 treatment is associated with the development of fulminant hepatitis, while the vice versa does not seem to be true [69].

## 6. Special Populations

### 6.1. Viral Diseases

Viral hepatitis infection or reactivation is one of the main exclusion criteria for IMH. ICIs may induce HBV reactivation through several mechanisms. Patients with chronic HBV infection have a pool of exhausted T-cells and ICIs may disrupt this balance and lead to its reactivation [70]. Most of the clinical trials about ICI therapy for cancer excluded patients with hepatitis C or B (HCV or HBV) and HIV infections (except for patients with hepatocellular carcinoma (HCC)). Due to this exclusion, the safety of ICIs among this population remains unclear. In a phase 1/2, open-label, non-comparative, dose escalation and expansion trial (CheckMate 040), patients with histologically confirmed advanced hepatocellular carcinoma with or without HBV and/or HCV infection were eligible to enroll. In both groups, no patient presented HBV reactivation during nivolumab, and few patients demonstrated transient reductions in HCV RNA levels [71]. Zhang and colleagues evaluated prospectively 129 HBsAg-positive cancer patients after PD-1 blockade and demonstrated that the rate of HBV reactivation was 21% (5/24) among subjects with undetectable HBV DNA and without antiviral therapy [72]. We have even less data on reactivations in patients with occult hepatitis B (OBI). OBI is defined as the presence of episomal covalently closed circular DNA molecule (cccDNA) in liver cells and/or HBV DNA in peripheral blood and hepatitis B surface antigen (HbsAg) negativity [73]. In our retrospective cohort study, we evaluated 150 potential-OBI (pOBI) patients (HbsAg negative) without HBV prophylaxis during immunotherapy. Only one pOBI patient (anti-Hbc+, anti-Hbs+ and anti-Hbe−) showed an increase in ALT (>3× ULN) with concomitant transient detectability of HBV-DNA [74]. Lin and colleagues demonstrated that seropositive HBsAg (OR = 6.30, *p* = 0.020), the existence of liver involvement (OR = 2.10; *p* = 0.030), and detectable baseline HBV DNA levels (OR = 2.39, *p* = 0.012) were risk factors for any-grade hepatotoxicity during ICIs, while the prophylactic antiviral therapy decreased hazard for the incidence of grade 3–4 hepatotoxicity (OR = 0.10, *p* = 0.016) [75].

Ravi and colleagues reported a similar rate of hepatotoxicity in patients HCV positive compared to the general population during treatment with ipilimumab [76]. Sangro et al. confirmed the good safety profile of tremelimumab in patients with HCC and chronic HCV infection, with only transient elevation of transaminases after the first dose [77]. 

HBsAg-positive patients should receive HBV antiviral therapy for at least 6 months after the end of ICI treatment. Antiviral therapy is not required for patients with HCV infection, but HCV replication should be monitored regularly [78].

In CheckMate 817, a phase 3B study, patients with metastatic NSCLC received flat-dose nivolumab plus weight-based ipilimumab. The enrolled patients could have an Eastern Cooperative Oncology Group (ECOG) performance status (PS) of 0-1-2, untreated brain metastases, renal impairment, hepatic impairment, or controlled HIV infection. IMH was reported in 3.5% of patients [79]. The experts stated that ICIs should be administered in people living with HIV (PLWH) when the viral load is undetectable and in patients receiving highly active antiretroviral therapy (HAART) CD4+; T-cell counts should be above 200 cells per mm3 [80].

### 6.2. Liver Transplant Recipients

There is very little information about the safety and efficacy of ICIs among organ transplant recipients. A pooled analysis of published cases in this special population demonstrated a considerably high risk of rejection and allograft loss [81], the risks and potential benefits of immunotherapy must therefore be carefully weighed up.

## 7. Conclusions

IMH is a not-so-rare complication of ICIs and it is extremely heterogeneous in its clinical presentation and severity. We currently require specific diagnostic tools and a better classification of toxicity grades. Also, the treatment of IMH is challenging, so a multidisciplinary team is useful to better manage this toxicity. A successful example of the important role of the multidisciplinary team is the ImmunoTOX evaluation committee. Established in 2016 at Gustave Roussy in France, it comprises an academic and multidisciplinary group of oncologists and organ specialists who employ a real-world, case-by-case approach to managing patients with irAEs [82]. This virtuous example of a multidisciplinary team could also be extended to other hospitals and other countries.

## Figures and Tables

**Table 1 cancers-16-00795-t001:** Grading assessment of immune-mediated hepatitis according to the Common Terminology Criteria of Adverse Events (CTCAE) and Drug-Induced Liver Injury Network (DILIN) criteria.

DILIN	CTCAE	Grade
Elevated serum ALT and/or ALP; TBil < 2.5 mg/dL; INR < 1.5; with or without symptoms (fatigue, weakness, nausea, anorexia, right upper abdominal pain, jaundice, pruritus, rash, or weight loss)	ALT/AST < 3× ULN; ALP/GGT > 1–2.5× ULN; TBili < 1.5× ULN	1
Elevated serum ALT and/or ALP; TBil ≥ 2.5 mg/dL or INR ≥ 1.5 without elevated TBil; symptoms may be aggravated	AST/ALT 3–5× ULN; ALP/GGT > 2.5–5× ULN; TBili 2–3× ULN	2
Elevated serum ALT and/or ALP; TBil ≥ 5 mg/dL with or without INR ≥ 1.5; symptoms are further aggravated; indication for hospitalization or prolonged hospitalization	AST/ALT 5–20× ULN; ALP/GGT > 5–20× ULN; TBili > 3× ULN	3
Elevated serum ALT and/or ALP; TBil ≥ 10 mg/dL or daily elevation ≥ 1.0 mg/dL; INR ≥ 1.5 with ascites, encephalopathy, or other organ dysfunction	AST/ALT > 20× ULN; ALP/GGT > 20× ULN; TBili > 10× ULN	4
Death	Death due to hepatoto×icity	5

ALT, alanine aminotransferase; AST, aspartate aminotransferase; ALP, alkaline phosphatase; TBil, total bilirubin; INR, international normalized ratio; ULN, upper limit of normal.

**Table 2 cancers-16-00795-t002:** Management algorithm for IMH.

Continue ICI; Check LT 1–2 Times Week	Grade 1
Hold ICI; check LT every 3 days; consider liver biopsy; if no improvement start steroid therapy (0.5–1 mg/kg/day of prednisone)	Grade 2
Hold ICI; check LT every 2 days; consider liver biopsy; if no improvement start steroid therapy (1–2 mg/kg/day of prednisone)	Grade 3
Hold ICI; check LT every 1 day; consider liver biopsy; if no improvement start steroid therapy (2 mg/kg/day of prednisone)	Grade 4

ICI: immune checkpoint inhibitors; LT: liver test.

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
