# Peer review of "The ABC of Immune-Mediated Hepatitis during Immunotherapy in Patients with Cancer: From Pathogenesis to Multidisciplinary Management"

_cancers, 2024, doi:10.3390/cancers16040795_

Round 1
Reviewer 1 Report
Comments and Suggestions for Authors
This is a review on recent evidence of immunotherapy related hepatoxicities. The review discusses the incidence, risk factors, possible pathological mechanisms, diagnosis and treatment. While there are some very good references, the manuscript is poorly written, and there are multiple sentences that are awkward, and have grammatical errors. It is also too casual. For example, one of sentences starts with "Anyways"
Some examples that come to mind include
1) line 91 - sentence that starts with "concerning the type of tumors..." I think the meaning is that when compared to patients without hepatic tumors, patients with HCC and underlying chronic liver disease are at higher risk of IMI WHEN COMPARED to patients without hepatic tumors. Instead they use "WITH" instead.
2) line 106-108 - this sentence is abrupt, and very awkward in phrasing
3) Please cite your references for line 102-103 - Patients with NAFLD are at higher risk of IMI than patients without. Please also be up to date with new terminology used such as MASLD.
Additionally, I am not sure the novelty of section "Histological diagnosis, General recommendations and When to start corticosteroids". In histological diagnosis, they are just citing the European oncology society recommendations. In the section about starting corticosteroids, they are just citing the definitions of grade 1-4 AEs.
Comments on the Quality of English Languageas above
Author Response
Response to reviewer 1: I am sorry that Reviewer 1 found the manuscript poorly written. The paper had been revised by a native English speaker who read his suggestions and made some small changes to the text.
Moreover, the requested bibliographic item is already written (23).
Regarding the novelty of the analyzed paragraphs, unfortunately, to date, there have been no substantial changes in the histological diagnosis of IMH.
Reviewer 2 Report
Comments and Suggestions for Authors
This manuscript addresses a complex but important arena that embraces a wide spectrum of disciplines; oncologists, hepatologists, internists, and emergency medicine physicians all of whom participate in the care of patients whom are undergoing immunotherapy. What has been your experience and feedback with other centers in countries who share similar facilities as yours?
Author Response
Response to reviewer 2: Thank you for your comments. We added the successful example of the ImmunoTOX evaluation committee.
Reviewer 3 Report
Comments and Suggestions for Authors
While this paper attempts to address immune-mediated hepatotoxicity (IMH) during immune checkpoint inhibitor (ICI) treatment, its methodology and presentation fall short of publish standards. The narrative lacks clarity in structuring the information and fails to offer substantial insights beyond well-known facts.
(1)what’s the differences of this paper between other published paper, sucah as: 10.3389/fphar.2022.1077468, Gastrointestinal and Hepatic Toxicities of Checkpoint Inhibitors: Algorithms for Management | American Society of Clinical Oncology Educational Book (ascopubs.org)
(2) Is hepatotoxicity caused by immunotherapy related to age, gender, lifestyle habits or medication history?
(3) Can the hepatotoxicity caused by it be fundamentally solved, such as the design and optimization of monoclonal antibodies, or optimizing the administration time and dose to reduce or eliminate this side effect? The author should give unique insights or opinions.
(4) The authors did not clearly provide case or literature sources, as well as case inclusion or exclusion criteria. This is indispensable for clinical research analysis.
Comments on the Quality of English LanguageMinor editing of English language required
Author Response
Response to reviewer 3: The cited paper discusses the main gastrointestinal toxicities from immunotherapy. Our review particularly focuses on IMH by reporting more recent literature data.
As reported in the paragraph 2.2: The reported incidence of immune-mediated hepatotoxicity (IMH) varies according to the class of ICI, types of tumors, underlying clinical conditions and medication history. The paragraph cites the most recent papers about this interesting topic.
Not being able to predict which patients will have IMH and which will not, it becomes difficult to simplistically assume a solution of treating patients in a different way.
As pointed out in the introductory paragraph, this is a narrative review and not a systematic literature review. We have chosen the papers most relevant to the topic of the review on the basis of our specific expertise.
Reviewer 4 Report
Comments and Suggestions for Authors
This narrative review addresses the timely and important issue of immune-mediated hepatitis caused by immune checkpoint inhibitor cancer therapies. It comprehensively summarizes current evidence on incidence, mechanisms, diagnosis, management, and outcomes of this emerging toxicity. The topic is relevant, the article is knowledgeable, and conclusions are supported, though some opportunities exist to strengthen the methodology. Overall, this review provides a useful overview of immune checkpoint inhibitor-associated hepatitis that could inform clinical practice after typical peer review refinements. Key remaining issues include systematically assessing study quality and better characterizing the strength of evidence for different aspects of this heterogenous condition.
However, there are several ways the authors could strengthen the manuscript: (if not currently feasible, please concisely clarify the following concerns)
Major Critiques:
- The review is described as "narrative" but there are opportunities to implement more systematic methods in the literature search, screening, data extraction, and evidence synthesis. Details on the search strategy, inclusion criteria, methods for data analysis/synthesis, and risk of bias assessments for individual studies should be provided.
- The actual risk of bias in the summarized evidence base is unclear. What is the quality of evidence for different aspects covered? Are there any low quality studies being given too much weight? A quality assessment should inform statements about evidence certainty.
- For some topics, quantitative synthesis (meta-analysis) may be possible if similar studies report compatible data. This would strengthen conclusions compared to qualitative summarization alone.
- There is no real consideration of limitations, inconsistencies, or evidence gaps in the data. A more critical look at the IMH evidence base should be provided.
Minor Suggestions and comments:
- The introduction could be condensed to allow more space for discussing evidence in the main content areas.
- Carefully check that statements about particular studies match with the actual data, interpretations, and limitations stated in those studies. Avoid overstating conclusions.
- Please highlight 2-3 of the most important unanswered research questions or evidence gaps in the conclusion. Additional ckeck-ups:
- The main question addressed is what is currently known about immune-mediated hepatitis (IMH) induced by immune checkpoint inhibitors (ICIs), covering topics ranging from pathogenesis to management.
- The topic is highly relevant and timely given the increasing use of ICIs in cancer treatment and the need to better understand potential immune-related adverse events like IMH. The paper provides a brief, narrative overview of this emerging issue.
- The paper summarizes current evidence on the incidence, risk factors, mechanisms, diagnosis, grading systems, histological features, and treatment approaches for IMH. It adds to the knowledge base by compiling data and recommendations that could inform clinical practice.
- The narrative format is appropriate for the broad scope. Specific suggestions to strengthen the methodology could include: a more systematic search and selection of literature, assessment of study quality, quantitative synthesis of results where feasible, and identification of key evidence gaps.
- The conclusions provide a reasonable summary of the presented evidence and relate back to the main objective of summarizing current knowledge on ICI-induced IMH.
- The references seem appropriate and directly support the content of the manuscript. Recent, relevant citations are included.
- The tables or figures should be expanded to avoid oversimplicity.
none.
Author Response
Response to reviewer 4: The above points refer to a systematic literature review with meta-analysis (PRISMA). It was not our intention to conduct a systematic review, but rather to provide a practical tool to have a general view of a problem that, although quite frequent, remains poorly managed in many cases.
The advice provided by the reviewer will be useful for a future in-depth study of risk factors for IMH.
The limitations of current knowledge are made explicit (if necessary) in the individual paragraphs, to provide the reader with a critical guide to the data.
The introductory paragraph is 20 lines in total, it has already been greatly summarised to leave more space for the main data.
Round 2
Reviewer 3 Report
Comments and Suggestions for Authors
This revised manuscript can be accepted with this version.
Comments on the Quality of English LanguageMinor editing of English language required
Reviewer 4 Report
Comments and Suggestions for Authors
This review provides an explicit update on the current landscape of ICI-associated hepatitis and it makes a timely contribution to an evolving clinical issue.
Comments on the Quality of English Languagenone.